# Effects of denitrification on the distributions of trace gas abundances in the polar regions: a comparison of WACCM with observations

Michael Weimer[1], Douglas E. Kinnison[2], Catherine Wilka[3], and Susan Solomon[1]

[1]Department of Earth, Atmospheric and Planetary Sciences, Massachusetts Institute of Technology, Cambridge, MA, USA
[2]Atmospheric Chemistry Observations & Modeling Laboratory, National Center for Atmospheric Research, Boulder, CO, USA
[3]Department of Earth System Science, Stanford University, Stanford, CA, USA

**Correspondence:** Michael Weimer (mweimer@mit.edu)

**Abstract.** Polar stratospheric clouds (PSCs) play a key role in the polar chemistry of the stratosphere. Nitric acid trihydrate (NAT) particles have been shown to lead to denitrification of the lower stratosphere. While the existence of large NAT particles (NAT "rocks") has been verified by many measurements especially in the Northern Hemisphere (NH), most current chemistry-climate models use simplified parameterizations, often based on evaluations in the Southern Hemisphere where the polar
vortex is stable enough that accounting for NAT rocks is not as important as in the NH. Here, we evaluate the probability density functions of various gaseous species in the polar vortex using one such model, the Whole Atmosphere Community Climate Model (WACCM), and compare these with measurements by the Michelson Interferometer for Passive Atmospheric Sounding onboard the Environmental Satellite (MIPAS/Envisat) and two ozonesonde stations for a range of years and in both hemispheres. Using the maximum difference between the distributions of MIPAS and WACCM as a measure of coherence, we
find better agreement for $HNO_3$ when reducing the NAT number density from the standard value of $10^{-2}$ used in this model to $5 \times 10^{-4}\,\mathrm{cm}^{-3}$ for almost all spring seasons during the MIPAS period in both hemispheres. The distributions of $ClONO_2$ and $O_3$ are not greatly affected by the NAT density. The average difference of WACCM to ozonesondes supports the need to reduce the NAT number density in the model. Therefore, this study suggests to use a NAT number density of $5 \times 10^{-4}\,\mathrm{cm}^{-3}$ for future simulations with WACCM.

## 1 Introduction

Polar stratospheric clouds (PSCs) have been known for decades to play a key role in explaining the stratospheric ozone hole (Solomon et al., 1986; Solomon, 1999; Tritscher et al., 2021). Reactions on their surfaces lead to activation of chlorine reservoirs, known as chlorine activation. In addition, sedimentation of PSCs result in dehydration and denitrification of the lower stratosphere. Three types of PSCs with different composition and roles have been found to be important: liquid supercooled
ternary solution droplets (STS, Carslaw et al., 1994) are major contributors to chlorine activation (e.g., Peter, 1997). Ice PSCs lead to dehydration of the stratosphere (e.g., Kelly et al., 1989). Nitric acid trihydrate particles (NAT) dominate the irreversible removal of nitric acid ($HNO_3$) from the lower stratosphere via sedimentation, known as denitrification, thus potentially reducing the reformation of chlorine reservoir species which can affect ozone loss (e.g., Waibel et al., 1999).

The existence of NAT particles in the stratosphere has been verified by a variety of airborne measurements (Fahey et al., 1989; Voigt et al., 2000; Fahey, 2001; Molleker et al., 2014; Woiwode et al., 2016). Since the polar vortex in the Southern Hemisphere (SH) generally is more stable than in the Northern Hemisphere (NH) (e.g., Schoeberl et al., 1992), the time period when denitrification can occur is much longer in the SH polar vortex than the NH. On the other hand, denitrification in the NH has been found to occur locally and the role of low-number density large-size NAT particles, so-called NAT rocks, has been discussed and verified by measurements (Fahey, 2001; Fueglistaler et al., 2002; Drdla and Müller, 2012; Adriani et al., 2004; Woiwode et al., 2014, 2016). These large particles can lead to significant sedimentation even on the shorter time scales needed to explain the occurrence of low $HNO_3$ volume mixing ratios (VMR) in the NH spring.

Satellite instruments are able to provide measurements of PSCs and gaseous $HNO_3$ with daily near-global coverage (Höpfner et al., 2018; Spang et al., 2018; Pitts et al., 2018; Santee et al., 2007; Wespes et al., 2022). Höpfner et al. (2006) found the first evidence of a NAT belt around the Antarctic Continent using the Michelson Interferometer for Passive Atmospheric Sounding (MIPAS) onboard the Environmental Satellite. Therefore, satellite measurements provide a unique opportunity for comparisons of $HNO_3$ and denitrification with current chemistry-climate models in both hemispheres. They also allow study of the range of the probability distributions in the data for different years and conditions.

Accounting for denitrification is an important process in chemistry-climate and in chemistry-transport models, which is why many parameterizations with different levels of detail have been developed to account for the microphysics and sedimentation of NAT particles in these models (e.g., Considine et al., 2000; Carslaw et al., 2002; Grooß et al., 2005; Wohltmann et al., 2010; Wegner et al., 2013; Zhu et al., 2015; Kirner et al., 2011; Weimer et al., 2021). Some models account for comprehensive microphysics of NAT particles (e.g., Zhu et al., 2015) to determine radii and concentrations which then redistribute gaseous $HNO_3$. Others use an intermediate approach accounting not for the full microphysics but constraining the amount of NAT to a measured NAT distribution (e.g., Kirner et al., 2011; Weimer et al., 2021). In addition, there are models with diagnostic approaches, as suggested e.g. by Considine et al. (2000), where NAT is formed based on the available gaseous $HNO_3$, sedimentation is computed and the $HNO_3$ contained in NAT particles is released back to the gaseous phase within the same model time step.

The standard version of the Whole Atmosphere component of the Community Earth System Model includes a diagnostic parameterization of NAT with a NAT number density set to a global value of $10^{-2}\,\mathrm{cm}^{-3}$ (Wegner et al., 2013). However, Wilka et al. (2021) found that this value leads to an overestimation of gaseous $HNO_3$ in the NH for the extraordinarily cold NH winter 2019/2020 in comparison to measurements of the Microwave Limb Sounder. They found a better agreement with the measurements using an adopted NAT number density of $N_{\mathrm{NAT}} = 10^{-5}\,\mathrm{cm}^{-3}$. Here, we investigate this further by applying an adapted version of the method by Zambri et al. (2021) to compare the distributions of various gaseous species within the polar vortex with MIPAS, which provided measurements during 2002 to 2012. MIPAS data is well suited to this task, because it provides high signal to noise local measurements over multiple years in both hemispheres, with good dynamic range; other datasets may also be appropriate but here we use MIPAS for this initial test. Further, using MIPAS and varying the NAT number density of the model, we can also investigate its associated impact on $HNO_3$, $ClONO_2$ and $O_3$ for many spring seasons in both hemispheres.

**Table 1.** Simulations in this study. In the "noHetAll" simulation, the heterogeneous reactions are removed apart from the reaction $N_2O_5 + H_2O$.

| Simulation | with heterogeneous chemistry? | $N_{NAT}$ (cm$^{-3}$) |
|---|---|---|
| HetAll.1e-2 | Yes | $10^{-2}$ |
| noHetAll.5e-4 | No | $5 \times 10^{-4}$ |
| HetAll.5e-4 | Yes | $5 \times 10^{-4}$ |
| HetAll.1e-5 | Yes | $10^{-5}$ |

Details about the model and the observation as well as the methods to compare them can be found in Sect. 2. Section 3 shows the comparison between the datasets. Finally, some concluding remarks are given in Sect. 4.

## 2 Datasets

### 2.1 WACCM

In this study, we use the Whole Atmosphere component (WACCM6) of the Community Earth System Model (CESM2.1) in Specified Dynamics (SD) mode (Gettelman et al., 2019; Danabasoglu et al., 2020) to compare distributions of the trace gases with the satellite measurements. The model is relaxed towards the Modern-Era Retrospective Analysis for Research and Applications version 2 (MERRA2, Gelaro et al., 2017). In this study, we use a horizontal resolution of $1.9°\,\mathrm{latitude} \times 2.5°\mathrm{longitude}$ and 88 vertical levels up to about $140\,\mathrm{km}$. In the component set "FWmaSD", a comprehensive chemistry for the middle atmosphere is included, as also used e.g. by Zambri et al. (2021).

Polar stratospheric clouds in WACCM are calculated using a diagnostic parameterization described by Considine et al. (2000), Kinnison et al. (2007) and Wegner et al. (2013). NAT particles are formed in thermodynamical equilibrium with the gaseous $H_2O$ and $HNO_3$ below the NAT threshold temperature (Hanson and Mauersberger, 1988). Coexistence of NAT and STS is accounted for by allowing $20\%$ of $HNO_3$ to form NAT whereas the rest is available for STS (Wegner et al., 2013; Solomon et al., 2015). Sedimentation of the NAT particles, i.e. the vertical redistribution of gaseous $HNO_3$, is calculated using a simple upwind scheme (Considine et al., 2000). The radius of the particles is determined in this scheme by using the amount of condensed $HNO_3$ and the NAT number density, which is a global parameter in this scheme. Larger NAT density leads to smaller particles and vice versa, assuming the mass of $HNO_3$ condensed in NAT is constant in the grid box. Therefore, the NAT density is a parameter in this scheme that can be used to tune the denitrification in the model. Examination of observations of NAT particle number densities (Pitts et al., 2009, 2011) and an emphasis on $HNO_3$ data for the SH in 2005 led Wegner et al. (2013) to adopt a NAT value of $0.01\,\mathrm{cm}^{-3}$. Fahey (2001) measured NAT number densities between $2.3 \times 10^{-4}$ and $2 \times 10^{-3}\,\mathrm{cm}^{-3}$. Pitts et al. (2009) and Pitts et al. (2011) used a wide range of $N_{NAT}$ between $10^{-4}$ and $10^{-1}\,\mathrm{cm}^{-3}$ based on in-situ measurements. In mountain waves, high NAT densities larger than $10^{-2}\,\mathrm{cm}^{-3}$ have been measured (e.g., Carslaw et al.,

1998). Voigt et al. (2005) observed a NAT PSC with $10^{-4}\,\mathrm{cm}^{-3}$. Here we have had the benefit of the unusual NH year 2020 and have placed more emphasis on both hemispheres to derive a parameterization that better represents the remaining gas phase $HNO_3$ in the maximum number of years and for both hemispheres. However, it is important to emphasize that the model's NAT parameterization is subject to multiple simplifications of complex microphysics. For instance, NAT particles in a grid box with a horizontal extent of about $100 \times 100\,\mathrm{km}^2$ do not necessarily have the same radius but usually follow a multi-modal size distribution (e.g., Fahey, 2001). In addition, NAT particles are not allowed to change their size over time or to be transported while interacting with the atmosphere by nucleation and (re-)sublimation and not all of the supersaturated amount of gaseous $HNO_3$ will become NAT. Therefore, observed NAT particle abundances may not be the best guide for this parameter choice.

We performed sensitivity simulations varying the NAT number density within the observed range from $10^{-2}$ to $10^{-5}\,\mathrm{cm}^{-3}$ in the current version 6 of WACCM. We also performed a simulation excluding stratospheric heterogeneous reactions, apart from the reaction $N_2O_5 + H_2O$, to evaluate the impact of heterogeneous processes on the gas-phase species. The $N_2O_5 + H_2O$ reaction rate is nearly independent of temperature, happens also on the background aerosols, does not directly affect the halogen chemistry and is important for the partitioning of reactive nitrogen in the atmosphere which is why we kept this reaction in that simulation. Without this reaction, the chemistry in the model would be completely unrealistic. All simulations are summarized in Table 1. They cover the satellite period starting from 1979 to present.

## 2.2 MIPAS

The Michelson Interferometer for Passive Atmospheric Sounding (MIPAS) operated in limb geometry on board the Environmental Satellite (Envisat) between July 2002 and April 2012 (Fischer et al., 2008). Envisat was placed in a sun-synchronous polar orbit at an altitude of around $800\,\mathrm{km}$ with more than 14 orbits per day. MIPAS measured a variety of trace gases including $HNO_3$, $ClONO_2$ and $O_3$ using a Fourier transform spectrometer in the infrared spectral range between 4.15 and $14.6\,\mathrm{\mu m}$ at tangent altitudes from 7 to $72\,\mathrm{km}$ (Fischer et al., 2008). The spatial resolution was approximately $3\,\mathrm{km}$ in the vertical and $30\,\mathrm{km}$ in the horizontal. The spectral resolution was $0.05\,\mathrm{cm}^{-1}$ between 2002 and March 2004. Due to technical issues with the satellite and a corresponding gap of measurements, MIPAS started measuring again in January 2005 with a reduced spectral resolution of $0.12\,\mathrm{cm}^{-1}$.

For this study, we use the V8 level 2 MIPAS retrievals from the IMKIAA processor (Kiefer et al., 2021, 2022; Von Clarmann et al., 2009) of $HNO_3$, $ClONO_2$ and $O_3$ to compare their distributions with the WACCM simulations. The reported precision in the region between 30 and $150\,\mathrm{hPa}$ is $5 - 15\,\%$ (around $100\,\mathrm{pptv}$) for $HNO_3$ (Sheese et al., 2016, 2017), $10\,\%$ (around $150\,\mathrm{ppbv}$) for $O_3$ (Sheese et al., 2017; Kiefer et al., 2022) and $7 - 32\,\%$ ($24 - 89\,\mathrm{pptv}$) for $ClONO_2$ (Sheese et al., 2016; Höpfner et al., 2007).

The method used to compare the MIPAS data with WACCM is based on the approach by Zambri et al. (2021). It takes advantage of WACCM's ability to provide output at the locations and times closest to the MIPAS profiles. In combination with interpolation of the MIPAS vertical levels to the WACCM altitudes, this enables a direct comparison of the two datasets. As a result, probability density functions (PDFs) can be evaluated in the spatiotemporal range of interest and compared to the observations.

Since NAT formation and denitrification are strongly temperature-dependent and most efficient at the lowest temperatures, we compute PDFs for profiles inside the polar vortex, determined by MERRA2 using the Nash criterion (Nash et al., 1996). As the sedimentation of the NAT particles takes several weeks (Tabazadeh et al., 2001), the largest effects of denitrification can be expected to be most amplified after the local winter, which is why we restrict our analysis to the early local spring, i.e.

1 February to 15 March in the Northern Hemisphere and 1 September to 15 October in the Southern Hemisphere, and to the pressure range 30 to $150\,\mathrm{hPa}$. Sensitivity studies by changing these two ranges did not affect the main message of the results shown in the next section. In order to make the datasets comparable, we remove the profiles from both WACCM and MIPAS data where negative values occur in the measurements. This does not have a significant effect on the results. The number of profiles depends on the area of the polar vortex and differs from year to year. Generally, the number of profiles is larger in the

SH due to the larger size of the SH polar vortex compared to the NH.

Zambri et al. (2021) used the Kolomogorov-Smirnov test to evaluate the difference in the PDFs between the model and observations, which uses the number of data points and the maximum difference in the cumulative density functions (CDFs) to check whether the distributions are distinguishable in a statistical sense related to a significance level $\alpha$. Here, we use the maximum difference in the CDFs $F$ of MIPAS and WACCM "max(d)" of the respective WACCM simulation to MIPAS directly

to evaluate the difference in the distributions (Zambri et al., 2021):

$$\mathrm{max(d)} = \max_x |F_{\mathrm{MIPAS}}(x) - F_{\mathrm{WACCM}}(x)| \tag{1}$$

with $x$ being the binned trace gas VMR. A more detailed discussion can be found in Zambri et al. (2021).

## 2.3   Ozonesondes

We also compare the WACCM simulations to balloon-borne in-situ measurements of ozonesondes, made available by the

World Ozone and Ultraviolet Data Centre. They use an electrochemical concentration cell to measure ozone profiles with a precision of 3 to $5\,\%$ and an uncertainty of about $\pm 10\,\%$ in the pressure range of interest here (Smit et al., 2007). We use ozonesondes of two stations: Eureka in the NH ($80.04\,°\mathrm{N}$, $86.18\,°\mathrm{W}$) and Syowa in the SH ($69.00\,°\mathrm{S}$, $39.58\,°\mathrm{E}$).

The method to compare the WACCM simulations with the ozonesonde measurements is similar to Wilka et al. (2021): The data is compared to the daily averaged WACCM ozone concentration at the grid point closest to the respective station for

profiles within the polar vortex. Averages and spread of the differences between WACCM and the observations are evaluated.

## 3   Impact on the distribution of gas-phase species

In this section, we will analyze the influence of changing denitrification in WACCM on the trace gases $HNO_3$, $ClONO_2$ and $O_3$ during early spring in both hemispheres. Generally, these species are expected to be influenced by a changed NAT number density because it leads to a redistribution of $HNO_3$ (lower $HNO_3$ at high altitudes and higher $HNO_3$ at lower altitudes).

During early spring, $HNO_3$ is photolyzed and forms $NO_2$ which combines with ClO to form $ClONO_2$. ClO is responsible for part of the catalytic ozone depletion and deactivated by the reaction into $ClONO_2$.

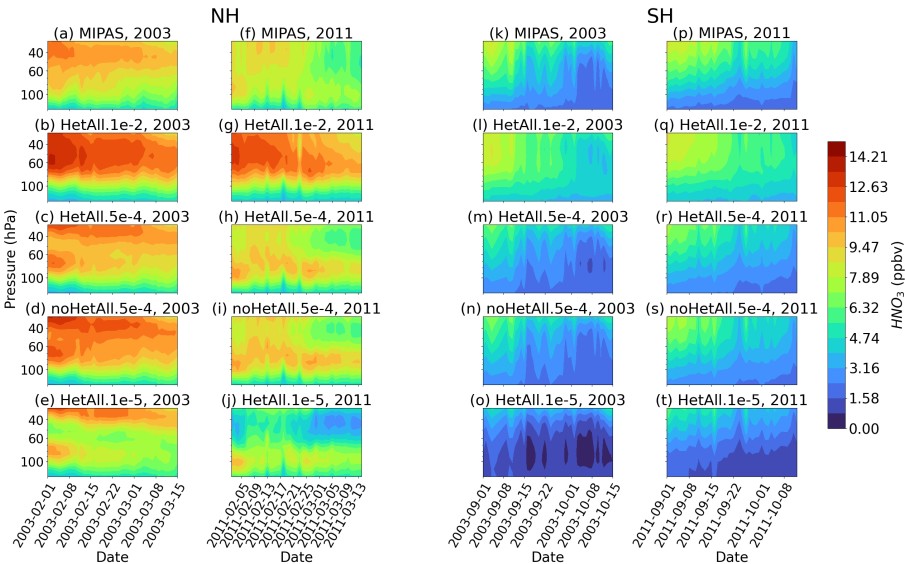

**Figure 1.** Daily vortex-averaged $HNO_3$ VMR profiles in the analyzed pressure range between 30 and 150 hPa during local spring for MIPAS (first row) and the WACCM simulations (other rows) in the NH (left two columns) and SH (right two columns) and for the years 2003 and 2011 (first and second column in the hemispheres). The number of profiles used to average each day depends on the availability of MIPAS data and the size of the polar vortex.

As a starting point, Fig. 1 shows vortex-mean profiles of the $HNO_3$ VMR during the local spring months for both hemispheres and the years 2003 and 2011 for MIPAS and all WACCM simulations of this study. Analysis of the other years of the MIPAS period showed similar results, which is why we focus on the two example years 2003 and 2011 here. The signature

of denitrification is indicated in the MIPAS measurements (first row) by $HNO_3$ VMR decreasing over time at around 30 to 60 hPa, with corresponding increases due to renitrification at the end of the shown timeseries at lower levels, like in 2011 in the NH (panel f). In the simulation with the current operational NAT number density of $10^{-2}\,\mathrm{cm}^{-3}$, HetAll.1e-2 in the second row of Fig. 1, the NAT particles do not get large enough for the simulated $HNO_3$ to compare well with the measured denitrification of the shown years. This means that larger $HNO_3$ VMRs than observed remain at higher altitudes at the end of the

timeseries, whereas the VMR is underestimated at lower altitudes. In contrast to this, with the smallest tested number density of $10^{-5}\,\mathrm{cm}^{-3}$, HetAll.1e-5 in the last row, the $HNO_3$ VMRs become too low, which indicates that the NAT particles become too large. By using a number density of $5 \times 10^{-4}\,\mathrm{cm}^{-3}$ (HetAll.5e-4 in the third row), the differences compared to MIPAS $HNO_3$ are smallest and the altitude regions of denitrification and renitrification are comparable to the measurements.

In order to generalize this comparison, Fig. 2 shows correlation plots of the HetAll simulations using different NAT number

densities compared to the MIPAS measurements for the whole pressure range $(30 - 150\,\mathrm{hPa})$ and all MIPAS years in the NH and SH. The simulation HetAll.5e-4 (green) has a relatively compact correlation with MIPAS in both hemispheres, whereas the other simulations show a larger scatter. The simulation HetAll.1e-2 shows $HNO_3$ VMRs that are too large compared to the observations because of NAT particles that are too small. As already suggested by Fig. 1, the $HNO_3$ VMRs in HetAll.1e-5 are

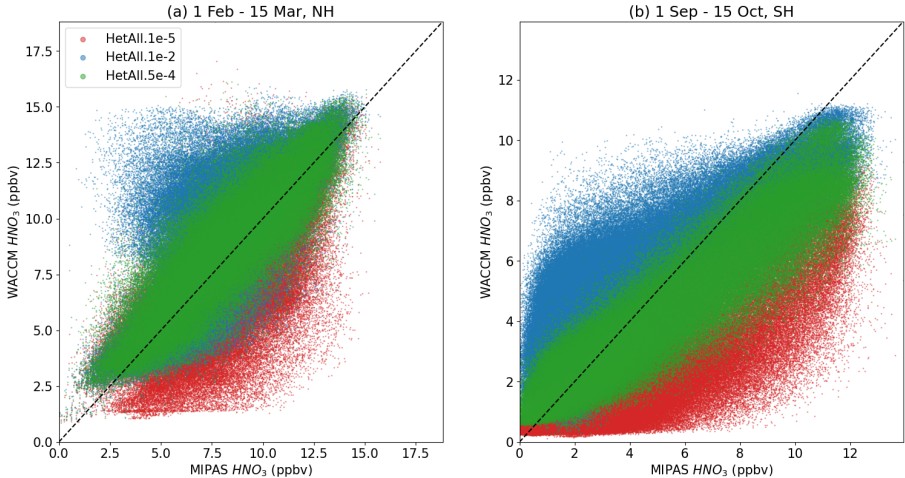

**Figure 2.** Scatter plots of MIPAS and WACCM $HNO_3$ in (a) the NH and (b) the SH for the HetAll simulation using the three NAT number densities described in the text. The dots include all measurement and model data between 30 and $150\,\mathrm{hPa}$. The dashed line corresponds to the one-by-one line.

too small compared to MIPAS as a result of too large NAT particles which lead to sedimentation below the shown pressure
range. As this figure includes the data of all MIPAS years and the whole pressure range, we will investigate the PDFs of this pressure range in the following, using the approach based on Zambri et al. (2021).

Probability density functions of $HNO_3$ for the whole MIPAS period 2002 to 2012 and for the years 2003 and 2011 are shown in Fig. 3 within the polar vortex in both hemispheres. All individual years can be found in the supplement of this study (Fig. S1). The blue line corresponds to the simulation with the largest NAT density, the red line shows the simulation with smallest
NAT density and the green and yellow lines show the simulations using $N_{\mathrm{NAT}} = 5 \times 10^{-4}\,\mathrm{cm}^{-3}$ , with the latter probing turning off all heterogeneous chemistry apart from $N_2O_5 + H_2O$. The black line corresponds to the MIPAS $HNO_3$ PDF.

Figure 3 demonstrates that the NAT density has a large impact on the PDFs in the model between 30 and $150\,\mathrm{hPa}$. In HetAll.1e-2, larger $HNO_3$ values are more frequent compared to the other simulations in all the panels. In contrast to this, HetAll.1e-5 underestimates $HNO_3$ as manifested by a higher frequency at smaller values. The simulation with peak values
closest to the MIPAS observations in all panels of Fig. 3 is HetAll.5e-4, indicating that heterogeneous chemistry is relevant and that $N_{\mathrm{NAT}} = 5 \times 10^{-4}\,\mathrm{cm}^{-3}$ leads to PDFs comparable to MIPAS for all years, see also Fig. S1 in the supplement. However, the NAT parameterization in WACCM is not able to capture the minimum and maximum values, especially in the SH where VMRs as high as $12.5\,\mathrm{ppbv}$ have been measured by MIPAS whereas the largest values in HetAll.5e-4 are around $10\,\mathrm{ppbv}$, e.g. in panel d of Fig. 3.

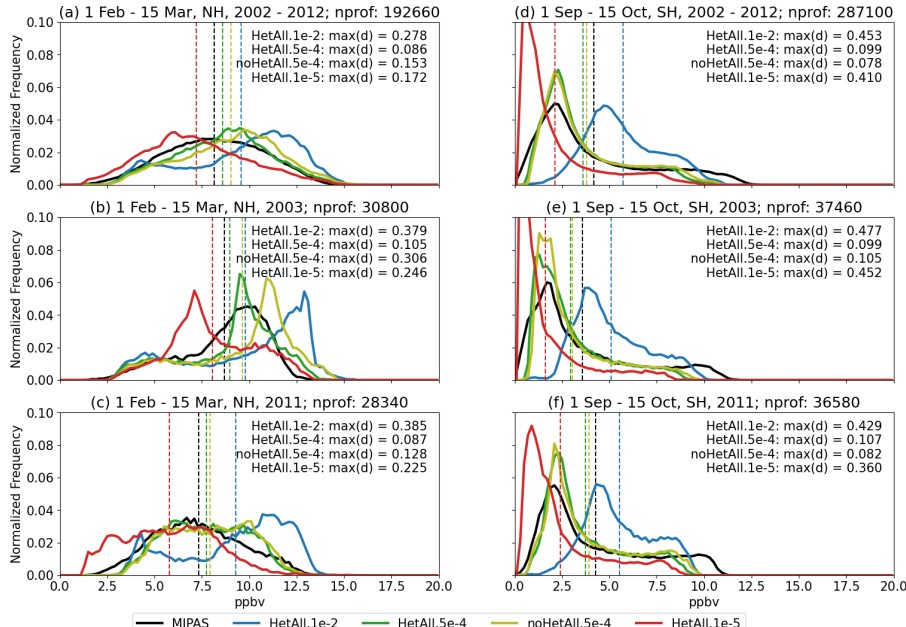

**Figure 3.** $HNO_3$ probability density functions within the polar vortex for MIPAS and the sensitivity simulations of WACCM for the NH (left column) and SH (right column) spring months in (a,d) the MIPAS period 2002 to 2012, (b,e) 2003 and (c,f) 2011. Vertical dashed lines show the average VMR of the respective simulations. The number of profiles used to derive the PDFs in each panel is denoted by "nprof". The maximum differences between the CDFs of MIPAS and the respective WACCM simulation are denoted by "max(d)" in the panels.

The overall agreement is also shown by the mean values illustrated by the dashed vertical lines in the figure. The mean $HNO_3$ value for HetAll.5e-4 is closest to the MIPAS mean value in all the panels, too. In addition, the maximum difference "max(d)" of the 5e-4 simulations is smallest in the shown years compared to the other simulations.

In almost all years, heterogeneous chemistry leads to about $0.5\,\mathrm{ppbv}$ smaller $HNO_3$ mixing ratios in the NH, see also Fig. S1 in the supplement. In the SH, this effect is not as large as in the NH. In total, the effect of halogen heterogeneous chemistry
on $HNO_3$ is small. Mean and shape of the distributions are similar for noHetAll.5e-4 and HetAll.5e-4.

Since the number of profiles differs from year to year depending on the development of the polar vortex, more weight in the distributions of the whole period 2002 to 2012 (panels a and d in Fig. 3) is given to years with a larger polar vortex. This is why the distributions of all individual years are shown in the supplement (Figs. S1 to S3). As a summary, the max(d) values of the individual years are shown in Tables 2 and 3 for NH and SH, respectively. The simulation with the minimum difference
is highlighted for each year. Apart from some exceptions in the NH, the minimum difference occurs for the simulations using $N_{\mathrm{NAT}} = 5 \times 10^{-4}\,\mathrm{cm}^{-3}$. This is another indication that this value improves denitrification and the distribution of $HNO_3$ in the polar vortex in both hemispheres.

Exceptions are the years 2004 and 2012 in the NH, where the simulation with the smallest NAT density (HetAll.1e-5) shows the minimum differences compared to the MIPAS distribution, see Table 2. In 2004, relatively warm temperatures in

**Table 2.** Maximum differences of WACCM to MIPAS $HNO_3$ CDFs ("max(d)" in the figures) for all years, 1 February to 15 March in the NH. Minimum values of each row are highlighted.

| Years | HetAll.1e-2 | HetAll.5e-4 | noHetAll.5e-4 | HetAll.1e-5 |
|---|---|---|---|---|
| 2002 – 2012 | 0.278 | **0.086** | 0.153 | 0.172 |
| 2003 | 0.379 | **0.105** | 0.306 | 0.246 |
| 2004 | 0.187 | 0.167 | 0.214 | **0.136** |
| 2005 | 0.283 | **0.119** | 0.190 | 0.296 |
| 2006 | 0.155 | **0.045** | 0.166 | 0.325 |
| 2007 | 0.297 | **0.194** | 0.302 | 0.220 |
| 2008 | 0.443 | **0.177** | 0.309 | 0.242 |
| 2009 | 0.148 | **0.095** | 0.193 | 0.141 |
| 2010 | 0.319 | **0.080** | 0.169 | 0.195 |
| 2011 | 0.385 | **0.087** | 0.128 | 0.225 |
| 2012 | 0.226 | 0.150 | 0.205 | **0.139** |

**Table 3.** Same as Table 2, but for the SH spring (1 September to 15 October).

| Years | HetAll.1e-2 | HetAll.5e-4 | noHetAll.5e-4 | HetAll.1e-5 |
|---|---|---|---|---|
| 2002 – 2012 | 0.453 | 0.099 | **0.078** | 0.410 |
| 2002 | 0.558 | 0.173 | **0.141** | 0.642 |
| 2003 | 0.477 | **0.099** | 0.105 | 0.452 |
| 2005 | 0.598 | **0.095** | 0.124 | 0.559 |
| 2006 | 0.553 | **0.141** | 0.210 | 0.366 |
| 2007 | 0.467 | 0.124 | **0.110** | 0.404 |
| 2008 | 0.495 | 0.112 | **0.089** | 0.432 |
| 2009 | 0.409 | 0.078 | **0.072** | 0.290 |
| 2010 | 0.419 | 0.156 | **0.129** | 0.425 |
| 2011 | 0.429 | 0.107 | **0.082** | 0.360 |

the NH stratosphere lead to small denitrification (Manney et al., 2005). As shown in Fig. 4a, this is reflected by the WACCM $HNO_3$ distributions which do not show large differences when changing the NAT density during that year. There seems to be a systematic difference between WACCM and MIPAS in this year, though, maybe due to a lower accuracy of the instrument or for instance due to a mismatch in the location of the polar vortex in the reanalysis data used for the model and the temperature history during that winter. The Arctic winter 2011/2012 was characterized by an early breakdown of the polar vortex at the end of December (Roy and Kuttippurath, 2022), but there is some evidence of large NAT particles during the cold period during

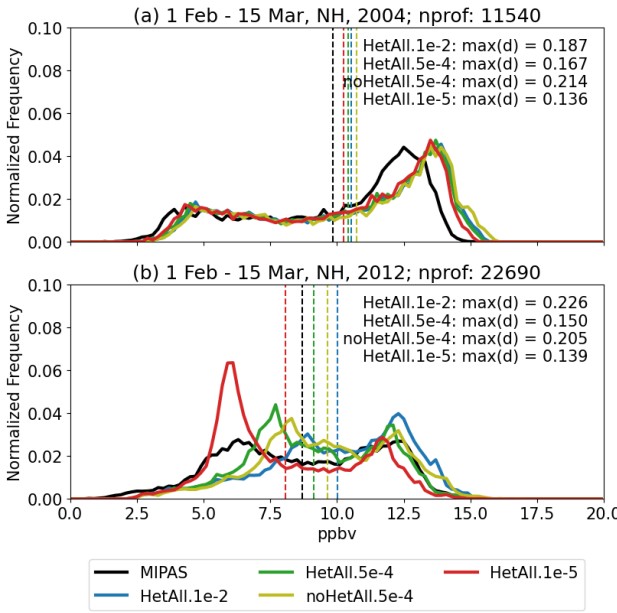

**Figure 4.** Same as Fig. 3, but for the years (a) 2004 and (b) 2012 in the NH.

December 2011 (Woiwode et al., 2019). This is why the $HNO_3$ distributions change in WACCM and probably large NAT particles play a larger role in determining the distributions for that year, see Fig. 4b.

Moderate changes due to the different NAT densities can also be found in other species. Figure 5 shows the distributions of chlorine nitrate ($ClONO_2$) for the same years and simulations as in Fig. 3. Both HetAll.1e-2 and HetAll.5e-4 have similar distributions and max(d) values. Depending on the year, minimum max(d) values occur in both simulations. HetAll.1e-5 seems to slightly underestimate the $ClONO_2$ mixing ratios in both hemispheres, illustrated by higher frequencies of low mixing ratios.

As a result of larger denitrification, impacts on the maximum values in the $ClONO_2$ distributions can be seen in both hemispheres. While the maximum $ClONO_2$ mixing ratios in the NH are closer to the MIPAS distributions using $N_{NAT} = 10^{-5}\,cm^{-3}$, the HetAll.5e-4 simulation captures the maximum values in the SH distributions.

There is also a clear signature of heterogeneous chemistry in the $ClONO_2$ distributions. The deactivation reaction forming $ClONO_2$

$$ClO + NO_2 + M \rightarrow ClONO_2 + M \tag{R1}$$

is usually faster than the deactivation reaction forming HCl

$$Cl + CH_4 \rightarrow HCl + CH_3, \tag{R2}$$

as long as $NO_2$ is available and the ozone VMRs are large enough (e.g., Kawa et al., 1992; Prather and Jaffe, 1990). At the edge regions of the polar vortex, some of the chlorine originally in HCl at the start of the winter shows up in $ClONO_2$ at

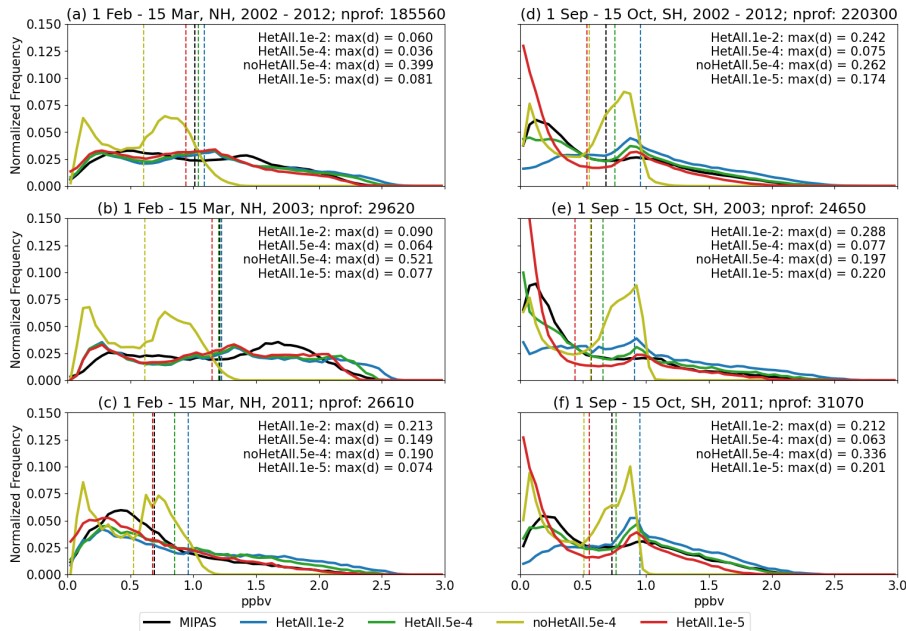

**Figure 5.** Same as Fig. 3, but for ClONO$_2$.

the end of the winter, known as the ClONO$_2$ "collar" (e.g., Toon et al., 1989; Douglass et al., 1995; Solomon et al., 2016). Therefore, while the simulation without heterogeneous chemistry noHetAll.5e-4 generally shows ClONO$_2$ VMRs that are too low compared to MIPAS, it is notable that the maximum values are increased by about 1 ppbv and then are comparable to MIPAS in all simulations with heterogeneous chemistry.

Similar impacts can be seen in the distributions of ozone (O$_3$), shown in Fig. 6. Ozone VMRs are about 1 ppmv too large in the SH for noHetAll.5e-4 compared to MIPAS. In the NH, the difference is smaller but ozone is still overestimated in the simulation without heterogeneous chemistry.

Wilka et al. (2021) showed that the agreement of WACCM model calculations of Arctic spring ozone losses in 2020 with ozonesonde data was heavily dependent on the denitrification parameterization used in the model, providing an important check. Figure 7 displays comparisons between WACCM and ozonesonde data at Syowa and Eureka for their respective spring seasons for all years between 2002-2012. A figure showing the comparison to Syowa without noHetAll.5e-4 can be found in the supplement (Fig. S4). Similar comparisons are provided for the South Pole in supplemental Fig. S5. Because local Antarctic ozone abundances can drop to values lower than 1 ppmv between about 15-20 km in years with large ozone losses (see e.g. Fig. 7a), care must be taken in interpreting percentage differences; we also provide absolute differences for that reason. The Syowa sondes agree considerably better with the model when the NAT number density of $5 \times 10^{-4}\,\mathrm{cm^{-3}}$ or less is used, compared to larger values of this parameter. At Eureka, results are far less sensitive due to smaller ozone losses, except in very cold years (like 2020 shown in Wilka et al., 2021).

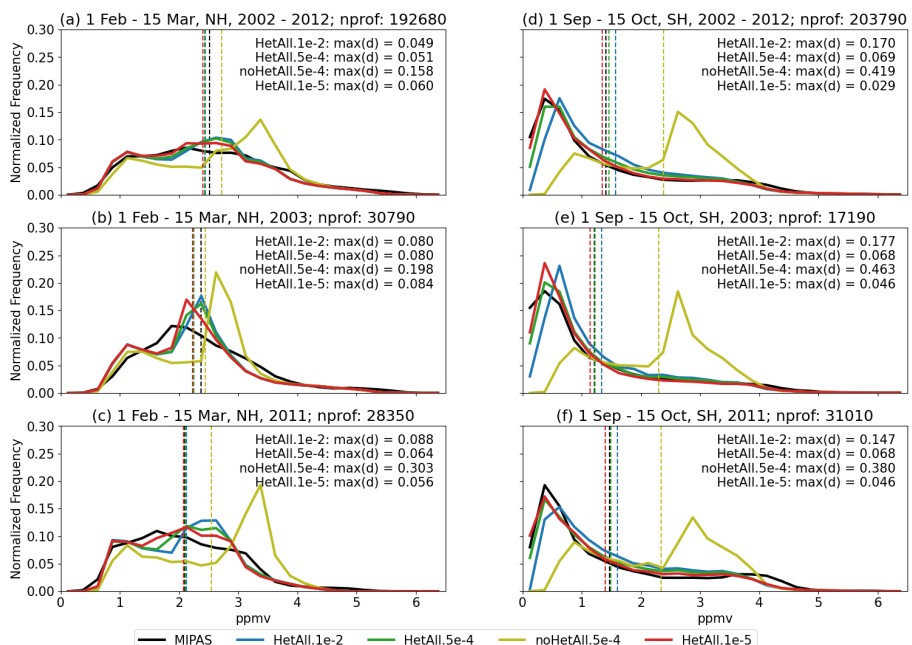

**Figure 6.** Same as Fig. 3, but for $O_3$.

In summary, the results in this section showed that the diagnostic parameterization of NAT particles in WACCM is able to reflect the main features in the three key gas-phase species $HNO_3$, $ClONO_2$ and ozone. When using a reduced NAT particle number density of $N_{NAT} = 5 \times 10^{-4}\,cm^{-3}$ compared to the standard WACCM setup, the difference in the cumulative density functions is minimized for almost all spring months of the MIPAS period 2002 to 2012 and the bias between 30 and $150\,hPa$ in comparison to the ozonesondes is reduced. This indicates that $N_{NAT} = 5 \times 10^{-4}\,cm^{-3}$ improves the distribution of these species in the polar regions and that it should be used in future WACCM simulations.

## 4  Discussion and Conclusions

In this study, we investigated the impact of the NAT number density ($N_{NAT}$) on the distributions of key gaseous species in the WACCM model. We compared probability density functions within the polar vortex with the MIPAS satellite instrument which operated between 2002 and 2012. As a measure of the difference between the distributions of WACCM and MIPAS, we used the maximum difference, max(d), between the cumulative density functions. To identify the impact of NAT on the distributions of the species, we performed sensitivity simulations varying $N_{NAT}$ between $10^{-2}$ and $10^{-5}\,cm^{-3}$. In the method applied here to compare the satellite measurements with the model, negative values in the satellite retrievals are cut off, which is why it cannot be used for noisy data (such as $N_2O_5$ and $ClO$ in MIPAS) and we restricted the analysis to the gaseous species $HNO_3$, $ClONO_2$ and $O_3$.

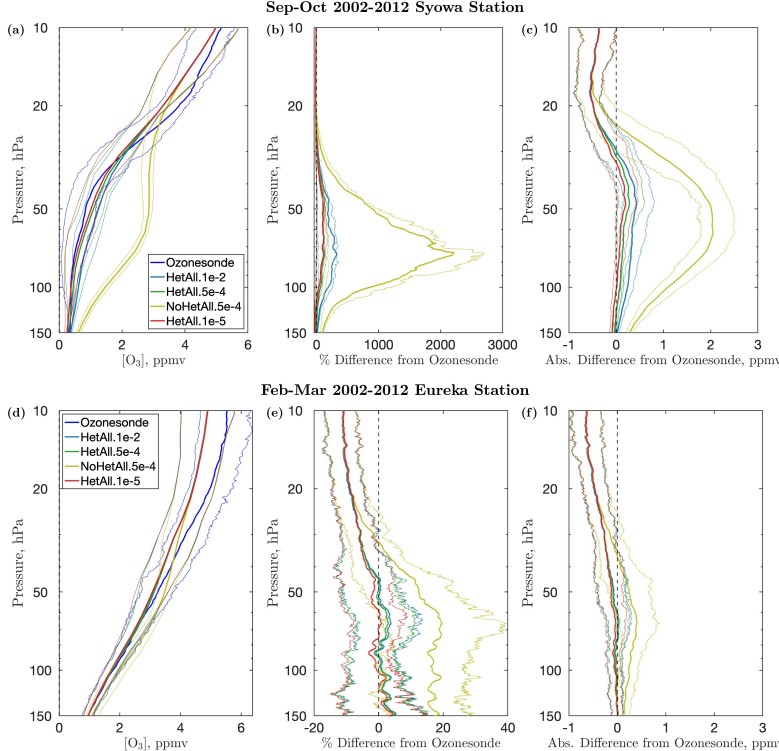

**Figure 7.** Mean and interquantile (0.25 and 0.75) range of (a,d) the volume mixing ratio and (b,e) relative differences and (c,f) the absolute differences between the respective WACCM simulations and two ozonesonde stations for profiles within the polar vortex using the same definition as in the previous analysis. The first row shows Syowa Station in the Antarctic and the second row shows Eureka Station in the Arctic. For WACCM, the grid point closest to the station is chosen. Average and distributions are taken for all profiles within the MIPAS period and the same months as for the satellite comparison. The left and right columns show the relative and absolute difference to the measurements, respectively.

The diagnostic parameterization of NAT in the WACCM model was shown to be able to reproduce the general shape of the MIPAS distributions for almost all years of the MIPAS period in the polar vortices of both hemispheres. There was a general overestimation of gaseous $HNO_3$, though, when using the standard $N_{NAT}$ of $10^{-2}\,cm^{-3}$. Reducing this number concentration to $5 \times 10^{-4}\,cm^{-3}$ was shown to also reduce the differences in the $HNO_3$ distributions between MIPAS and WACCM for almost all spring seasons in both hemispheres during the MIPAS period. Changes in the distributions of $ClONO_2$ and $O_3$ due to the new value of $N_{NAT}$ were either negligible or the differences were reduced. Mean differences between ozonesonde profiles and the WACCM simulations at the grid point closest to the sonde stations Syowa and Eureka were also shown to be decreased when reducing $N_{NAT}$. Therefore, this study suggests the use of $N_{NAT} = 5 \times 10^{-4}\,cm^{-3}$ for future simulations. This new value is within the range of measured NAT number densities.

A simplified parameterization as applied in WACCM cannot capture all features of the distributions of all years. This was demonstrated by the years 2004 and 2012 where differences between the MIPAS and WACCM distributions were increased. Further, there are many uncertainties in the detailed chemistry from one year to another that may also be important for denitrification and ozone loss, including for example volcanic particle inputs, gravity and planetary wave amplitudes, and changes in circulation, to name only a few. It can also be seen that the smallest number density of $10^{-5}\,\mathrm{cm}^{-3}$ is comparable to measurements during the exceptionally cold northern winter 2019/2020 (Wilka et al., 2021). In addition, although overall differences in the PDFs are reduced with $N_{\mathrm{NAT}} = 5 \times 10^{-4}\,\mathrm{cm}^{-3}$, the Kolmogorov-Smirnov test, used e.g. by Zambri et al. (2021) with a similar approach as in this study, will fail if not using a significance level $\alpha$ that is set to a very small value ($O(10^{-70})$). Therefore, although the differences are decreased, the distributions of MIPAS and WACCM are still different in a statistical sense, probably due to the simplifications in the NAT parameterization of WACCM.

Future studies could apply this methodology to other long-term satellite measurements to evaluate the usage of this new NAT number density for further years, although their precision has to be low enough to allow for comparisons with individual profiles. Nevertheless, as a follow-up of the publication by Wilka et al. (2021), we recommend using $N_{\mathrm{NAT}} = 5 \times 10^{-4}\,\mathrm{cm}^{-3}$ for future simulations with WACCM, indicating that NAT rocks play an important role, especially in the NH where we saw the largest impact on changing the NAT number density.

*Code and data availability.* Instructions for the access to MIPAS data can be found at https://www.imk-asf.kit.edu/english/308.php (last access on 18 November 2022). Instructions how to download CESM version 2 can be found in Danabasoglu et al. (2020). Ozonezonde station data are publicly available through the World Ozone and Ultraviolet Data Center (WOUDC; WMO/GAW Ozone Monitoring Community, 2022). The scripts and data needed to create the figures of this paper can be found at https://g-27eb33.7a577b.6fbd.data.globus.org/ACP_Weimer_2022/ACP_Weimer_2022.tar.gz (last access on 7 December 2022).

*Author contributions.* MW added the comparison with MIPAS and wrote the first draft of the manuscript. DEK performed the simulations with WACCM used for the comparisons. CW added the comparison to ozonesondes. All authors contributed to prepare the manuscript.

*Competing interests.* The authors declare that they have no competing interests.

*Acknowledgements.* Michael Weimer and Douglas E. Kinnison were funded in part by NASA grant 80NSSC19K0952. Susan Solomon was funded in part by grant no. 1906719 of the U.S. National Science Foundation (NSF). We would like to thank Brian Zambri for initiating this work and helping with the comparison to MIPAS. This material is based upon work supported by National Center for Atmospheric Research (NCAR), which is a major facility sponsored by NSF under the Cooperative Agreement 1852977. The CESM project is supported primarily by NSF. We would like to acknowledge support from the Svante cluster at MIT used for the comparison of model and the data,

see https://svante.mit.edu/intro.html (last access 18 November 2022) for more information. We would like to acknowledge high-performance computing support from Cheyenne and Casper provided by NCAR's Computational and Information Systems Laboratory (2019), sponsored by NSF.

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
