# Peer review of "Effects of denitrification on the distributions of trace gas abundances in the polar regions: a comparison of WACCM with observations"

_EGUsphere, 2022_

## Referee Comment (RC1)

**General comments**

This article is about improving the parameterization of denitrification in the WACCM model with the help of MIPAS measurement data.

The paper is scientifically sound, well written and well structured. I recommend to publish the study with (relatively) minor revisions.

Having said this, I think I have two major comments. I have put all my comments into the "specific comments" section.

The first one is the comment to lines 90–93 (see below). I think you are doing a little bit of a strategy of "least effort" here. I think some more discussion and analysis would really be helpful and increase insight and maybe even improve your parameterization.

The second major comment is to lines 165–167 (see below) and Figure 4. This one really worries me. The ozone values are off by an order of magnitude here. I hope that there this some simple explanation for this, e.g. a simple mistake in the labels of the x axis. Or maybe I am just too tired after doing this review in one piece in only 6 hours, and can't think clearly anymore ;-) .

While this paper is quite technical, the improvement of model parameterizations is clearly an important topic, and as such the manuscript is well suited to ACP. Having said this, it would also be an obvious candidate for publication in GMD, but I have no strong opinion on the journal this article should be published in.

Best wishes, Ingo Wohltmann

**Specific comments**

- Title: This manuscript is specifically about the WACCM model. This should also be reflected in the title. Please add "WACCM" to the title in some way. For the same reason, you could also add that you use MIPAS data in the title.

- Lines 38–40: "Accounting for denitrification is an important process in chemistry-climate models, which is why many parametrizations with different levels of detail have been developed to account for the microphysics and sedimentation of NAT particles in these models (e.g., Wegner et al., 2013; Zhu et al., 2015; Kirner et al., 2011; Weimer et al., 2021)."

  I am a little bit surprised that you are talking only about chemistry-climate models (CCMs) here, and not about chemistry and transport models (CTMs). This is all the more surprising since you use the specified dynamics version of WACCM here (SD-WACCM), which is effectively a CTM.

  There are some obvious omissions in the citations that you give here. The first that comes to my mind is the implementation of denitrification into the CLaMS model by Grooß et al., doi:10.5194/acp-5-1437-2005 (there are also follow-up papers), which is based on the Lagrangian DLAPSE

scheme (Carslaw et al., doi:10.1029/2001JD000467) that has also been used in other models.

I probably have a small conflict of interest here, but if you like, you could also cite the implementation of DLAPSE into my model ATLAS, Wohltmann et al., doi:10.5194/gmd-3-585-2010.

But surprisingly, you also don't cite your own implementation. As far as I can figure out, the denitrification scheme in WACCM is based on the scheme described in Considine et al. (2000). Wouldn't it make sense to cite that here? Wegner et al. does not exactly deal with the details of the denitrification scheme.

- Same topic: I would really like to see a short and concise account of the methods used to model denitrification in CCMs and CTMs here. This really is not required to be detailed or complicated, just a short overview. I think that would be helpful for the reader.

- Line 48: "...other datasets may also be appropriate but here but here we use MIPAS for this initial test".

  You probably anticipated a comment here and tried to avoid it by stating this. Nevertheless, I will give the comment here anyway ;-) . There are some obvious additional candidates for comparison with measurements. The most obvious one is MLS, in particular since there is a much longer time series extending into the present.

- Line 49 (and lines 11–12): If you would use other measurement sources than MIPAS, you could also show results for HCl here. HCl seems to be quite an obvious omission here.

- Line 64: "...using a simple upwind scheme."

  I would add Considine et al. as a reference at the end of the sentence, since this is where it is actually described in detail.

- Same topic: I think it wouldn't hurt to shortly describe how the denitrification scheme works, even if you repeat information from some older papers, since this is central for your study. I think this would help the reader, who is not required to gather this information from other papers then.

  This could give the reader an idea how microphysical processes as nucleation, sedimentation (fall velocities), growth and evaporation are handled here in a simplified fashion.

- Lines 67–68: "...observed NAT particle abundances may therefore not be the best guide for this parameter choice."

  I agree, and will therefore not comment on this or other details of PSC measurements.

- Line 69: "Here we have had the benefit of the unusual NH year 2020"

  Exactly. But since you use MIPAS data here, which ends in 2012, you are not able to compare to just this winter, which could be relatively easily compensated for by using a different measurement data set like MLS.

- Line 71: I think it would increase the insight of the reader in what is happening in your model greatly if you would mention that decreasing the NAT number density will increase the particle sizes and the fall velocities, and hence, the efficiency of denitrification. Maybe you mention that somewhere else in the paper and I missed it.

  This is under the caveat that I understand correctly how your denitrification scheme is working.

- Line 90–93, 99, General approach with pdfs in the polar vortex in the altitude range 30–150 hPa: I understand the benefit and elegance of your approach to have a single and comprehensive metric (integrated difference of the WACCM and MIPAS pdfs) that you apply for improving your parameterization, but I think this very condensed metric may hide some important things.

  In particular, 30–150 hPa is a HUGE altitude range, where all kinds of parameters change significantly (pressure, partitioning of species, NAT threshold temperatures and so on). Isn't it a bit dangerous to throw everything into one pdf here?

  I think it is required that you at least go into SOME more detail here. That would not only allow more insight into what is actually happening, but may also help to improve the parameterization.

  E.g., I would really like to see a comparison of MIPAS and WACCM using an HNO3 vortex mean profile that shows regions of denitrification and renitrification. This could also be done as a function of time and altitude for individual years.

  Or, since the two pdfs may agree although identical values are located at spatially very different locations (in altitude or horizontally), it would also be helpful to do some simple "sanity checking" and to provide some simple plots like contour plots of HNO3 at a given altitude level for several dates (in different years).

- Line 100: "In order to make the datasets comparable, we remove the profiles from both WACCM and MIPAS data where negative values occur in the measurements."

  Does this significantly affect the results? If not (what I suspect), it may be worth noting. I write this comment because removing profiles with negative values from a noisy measurement data set can easily introduce a positive bias when e.g. applying a mean or looking at the pdf.

- Line 115: Probably only when the ozone sonde measurement is inside the vortex? Might be worth noting.

- Line 118: Why these two years? Warm and cold NH winter?

- Line 126: But I think it also shows that compared to the effects of denitrification, the effects of heterogeneous chemistry on HNO3 are usually quite small. Might be worth noting. Would also be nice to have a citation for this, but since this is kind of textbook knowledge not often stated explicitly, it might be hard to find.

- Line 161: "There is also a clear signature of heterogeneous chemistry in the ClONO2 distributions."

  ...which I don't find surprising when you switch off the ClONO2+HCl activation reaction.

- Line 161: "ClONO2 is formed faster in early spring than HCl"

  I don't really know what to make out of this sentence. Do you want to discuss the fact here, that depending on temperatures and hemispheres, chlorine deactivation is sometimes predominantly into HCl (SH, maybe cold NH winters) and sometimes predominantly into ClONO2? That effect will depend on the Cl/ClO ratio and hence on low or high ozone, and it will depend on denitrification.

- Line 163–164: "ClONO2 volume mixing ratios that are too low compared to MIPAS, it is notable that the maximum values are increased by about 1 ppbv and then are comparable to MIPAS in all simulations with heterogeneous chemistry."

  That's interesting. Switching off heterogenous chemistry should impede activation of ClONO2 into active chlorine and impede ozone depletion. But why does this seem to "cut off" the highest ClONO2 values? Do you have an explanation? Seems not straightforward to me.

- Line 165–167: Now, I am confused. When switching off heterogenous chemistry, I would have expected MUCH higher ozone values (a few ppm). Am I misunderstanding something here completely or is there something going wrong here (e.g. the values at the x axis not being correct)?

  And the ozone values in the northern hemisphere seem to be suspiciously low in all cases. Values of below 0.3 ppm are normally never observed in this altitude range, even with ozone depletion.

  I have the impression that something is clearly going wrong here.

- Lines 168 and following: Why don't you compare to ozone from MIPAS here? Would somehow be consistent to the rest of the manuscript. I would also find it interesting not only to see the differences, but also the mean ozone profiles.

- Lines 207–208: Isn't it a little bit over the top to argue with the Kolmogorov-Smirnov test here when it can be easily seen from the figures that the distributions are still different? I think that could be shortened.

- Supplement: I think it wouldn't hurt to mention what species you are showing in the plots. In the moment, you can only deduce that from the figure numbers in the main manuscript given in the caption.

**Technical corrections**

- Line 124: Typo "cpmpared"

---

## Author Comment (AC1)

Dear Ingo Wohltmann,

Thank you very much for this detailed review of our manuscript. Your comments helped us to improve the manuscript and we hope that we could address all of your major concerns in the revised manuscript. Please find below our responses to each of your comments. Major changes include:

- We changed the title and included WACCM, as suggested in your first comment

- We added more information about the NAT parameterization in WACCM

- Some more information about the method how to calculate the max(d) values is provided in Sect. 2.2

- We added two new figures comparing MIPAS $HNO_3$ with WACCM as (1) a timeseries and (2) a scatter plot, see discussion below and in the revised manuscript

Thanks again for your review and on behalf of all authors,
Michael Weimer

**1   Specific Comments**

**1.1   Title: This manuscript is specifically about the WACCM model.  This should also be reflected in the title.  Please add "WACCM" to the title in some way. For the same reason, you could also add that you use MI-PAS data in the title.**

We agree that the manuscript is solely focussed on the WACCM model.  As we're using observations other than MIPAS, we suggest changing the title to "Effects of denitrification on the distributions of trace gas abundances in the polar regions: a comparison of WACCM with observations".

**1.2 Lines 38–40: "Accounting for denitrification is an important process in chemistry-climate models, which is why many parametrizations with different levels of detail have been developed to account for the microphysics and sedimentation of NAT particles in these models (e.g., Wegner et al., 2013; Zhu et al., 2015; Kirner et al., 2011; Weimer et al., 2021)." I am a little bit surprised that you are talking only about chemistry- climate models (CCMs) here, and not about chemistry and transport models (CTMs). This is all the more surprising since you use the specified dynamics version of WACCM here (SD-WACCM), which is effectively a CTM. There are some obvious omissions in the citations that you give here. The first that comes to my mind is the implementation of denitrification into the CLaMS model by Grooß et al., doi:10.5194/acp-5-1437-2005 (there are also follow-up papers), which is based on the Lagrangian DLAPSE scheme (Carslaw et al., doi:10.1029/2001JD000467) that has also been used in other models. I probably have a small conflict of interest here, but if you like, you could also cite the implementation of DLAPSE into my model ATLAS, Wohltmann et al., doi:10.5194/gmd-3-585-2010. But surprisingly, you also don't cite your own implementation. As far as I can figure out, the denitrification scheme in WACCM is based on the scheme described in Considine et al. (2000). Wouldn't it make sense to cite that here? Wegner et al. does not exactly deal with the details of the denitrification scheme.**

Thank you for this comment. We added some citations for CTMs and Considine et al. (2000) as well as others. Please note that WACCM is a chemistry-climate model which, in SD mode, is indeed depending on external meteorological data. The ratio between using MERRA2 and the internal WACCM meteorology in the configuration in this study is only 1/100, though, which means that the climatology is the same as in the reanalysis data but for shorter time scales it can be assumed to be free-running. Thus, we hesitate to consider WACCM as a CTM in any configuration.

**1.3 Same topic: I would really like to see a short and concise account of the methods used to model denitrification in CCMs and CTMs here. This really is not required to be detailed or complicated, just a short overview. I think that would be helpful for the reader.**

We agree with this comment and added some sentences in the introduction pointing out the different approaches to parameterize NAT in the models.

**1.4 Line 48: "... other datasets may also be appropriate but here we use MIPAS for this initial test". You probably anticipated a comment here and tried to avoid it by stating this. Nevertheless, I will give the comment here anyway ;-) . There are some obvious additional candidates for comparison with measurements. The most obvious one is MLS, in particular since there is a much longer time series extending into the present.**
– and –
Line 49 (and lines 11–12): If you would use other measurement sources than MIPAS, you could also show results for HCl here. HCl seems to be quite an obvious omission here.
– and –
Line 69: "Here we have had the benefit of the unusual NH year 2020" Exactly. But since you use MIPAS data here, which ends in 2012, you are not able to compare to just this winter, which could be relatively easily compensated for by using a different measurement data set like MLS.

We agree that it would be beneficial to compare the WACCM simulations with other datasets, such as MLS, which are still in operation and would allow for comparisons of more recent seasons than with MIPAS and other trace gases. Speaking about MLS, however, the signal-to-noise ratio is not large enough to do comparisons to other datasets without averaging, as found for HCl e.g. by Wang et al., PNAS (2023), doi:10.1073/pnas.2213910120, but this is true for all species measured by MLS. Here, we analyze individual profiles and compare WACCM and the satellite data at these locations, so averaging is not possible which is why we selected MIPAS for this study. We already mention at the end of the manuscript that analysis with other datasets would be beneficial and added another sentence about this discussion here.

**1.5  Line 64: "... using a simple upwind scheme." I would add Considine et al. as a reference at the end of the sentence, since this is where it is actually described in detail.**

We added this to the manuscript.

**1.6  Same topic: I think it wouldn't hurt to shortly describe how the denitrification scheme works, even if you repeat information from some older papers, since this is central for your study. I think this would help the reader, who is not required to gather this information from other papers then. This could give the reader an idea how microphysical processes as nucleation, sedimentation (fall velocities), growth and evaporation are handled here in a simplified fashion.**

We added some more details about the NAT scheme in WACCM to Sect. 2.1.

**1.7  Lines 67–68: ". . . observed NAT particle abundances may therefore not be the best guide for this parameter choice." I agree, and will therefore not comment on this or other details of PSC measurements.**

Thank you.

**1.8  Line 71: I think it would increase the insight of the reader in what is happening in your model greatly if you would mention that decreasing the NAT number density will increase the particle sizes and the fall velocities, and hence, the efficiency of denitrification. Maybe you mention that somewhere else in the paper and I missed it. This is under the caveat that I understand correctly how your denitrification scheme is working.**

We added this information also to Sect. 2.1.

**1.9  Line 90–93, 99, General approach with pdfs in the polar vortex in the altitude range 30–150 hPa:** I understand the benefit and elegance of your approach to have a single and comprehensive metric (integrated difference of the WACCM and MIPAS pdfs) that you apply for improving your parameterization, but I think this very condensed metric may hide some important things. In particular, 30–150 hPa is a HUGE altitude range, where all kinds of parameters change significantly (pressure, partitioning of species, NAT threshold temperatures and so on). Isn't it a bit dangerous to throw everything into one pdf here? I think it is required that you at least go into SOME more detail here. That would not only allow more insight into what is actually happening, but may also help to improve the parameterization. E.g., I would really like to see a comparison of MIPAS and WACCM using an HNO3 vortex mean profile that shows regions of denitrification and renitrification. This could also be done as a function of time and altitude for individual years. Or, since the two pdfs may agree although identical values are located at spatially very different locations (in altitude or horizontally), it would also be helpful to do some simple "sanity checking" and to provide some simple plots like contour plots of HNO3 at a given altitude level for several dates (in different years).

We are aware that the pressure range shown in the figures is quite large, but as we already mention in the manuscript, we also analyzed sub-regions of the pressure range 30 to 150 hPa with similar results. In addition, the number concentration of NAT in the model under investigation here is set to one global value, but might vary spatio-temporally in reality. To get a general overview how to set this parameter, we decided that showing the whole pressure range is appropriate for the purpose of this study. In the revised manuscript, we analyzed the height-dependent HNO3 in the vortex, which can be found in the new Fig. 1 showing that the height-dependent distribution is similar in the polar vortex, with smallest differences for HetAll.5e-4. We also added a scatter plot (new Fig. 2) including all profiles of all years showing that HetAll.5e-4 has a clear correlation to MIPAS whereas the simulations using different NAT number density have a larger spread. Please note that this includes the whole pressure range from 30 to 150 hPa. Therefore, smallest max(d) values for HetAll.5e-4 are

not just a coincidence, but they result from a better correlation at all altitudes, which supports our approach to use the whole pressure range in the PDFs.

**1.10 Line 100: "In order to make the datasets comparable, we remove the profiles from both WACCM and MIPAS data where negative values occur in the measurements." Does this significantly affect the results? If not (what I suspect), it may be worth noting. I write this comment because removing profiles with negative values from a noisy measurement data set can easily introduce a positive bias when e.g. applying a mean or looking at the pdf.**

Thank you for this comment. As we describe in the manuscript, output of the WACCM profiles is given at the locations closest to the MIPAS measurements. Since we remove these profiles from both WACCM and MIPAS, there should not be a systematic bias between them induced by that. We also added a sentence that this does not affect the results significantly.

**1.11 Line 115: Probably only when the ozone sonde measurement is inside the vortex? Might be worth noting.**

We excluded ozonesonde profiles outside the polar vortex from Fig. 7 in the revised manuscript and added this information to the methodology section 2.3 and to the figure caption.

**1.12 Line 118: Why these two years? Warm and cold NH winter?**

As can be seen in the supplement, all individual years show similar results in both hemispheres. Therefore, we showed two example years here. We added a reference to the supplement at this point in the manuscript and a short explanation at the start of Sect. 3 that these are example years.

**1.13 Line 126: But I think it also shows that compared to the effects of denitrification, the effects of heterogeneous chemistry on HNO3 are usually quite small. Might be worth noting. Would also be nice to have a citation for this, but since this is kind of textbook knowledge not often stated explicitly, it might be hard to find.**

We added a sentence to the next paragraph explicitly describing this.

**1.14 Line 161: "There is also a clear signature of heterogeneous chemistry in the ClONO2 distributions." ... which I don't find surprising when you switch off the ClONO2+HCl activation reaction.**

We agree with this comment, but we still think that it is worth noting. In addition, this is a nice example where the effect of heterogeneous chemistry can be clearly seen at the first glance.

**1.15 Line 161: "ClONO2 is formed faster in early spring than HCl" I don't really know what to make out of this sentence. Do you want to discuss the fact here, that depending on temperatures and hemispheres, chlorine deactivation is sometimes predominantly into HCl (SH, maybe cold NH winters) and sometimes predominantly into ClONO2? That effect will depend on the Cl/ClO ratio and hence on low or high ozone, and it will depend on denitrification.**
**– and –**
**Line 163–164: "ClONO2 volume mixing ratios that are too low compared to MIPAS, it is notable that the maximum values are increased by about 1 ppbv and then are comparable to MIPAS in all simulations with heterogeneous chemistry." That's interesting. Switching off heterogenous chemistry should impede activation of ClONO2 into active chlorine and impede ozone depletion. But why does this seem to "cut off" the highest ClONO2 values? Do you have an explanation? Seems not straightforward to me.**

We agree that this statement is not true in this generality as it occurs in the manuscript. What we wanted to emphasize here is that the reaction ClO + NO2

is usually faster than Cl + CH4, but only if NO2 is available and ozone VMRs are large enough (e.g. Solomon et al., JGR, 2015). The dependence on NO2 shows that this process depends on denitrification. At the edge regions of the polar vortex the well known ClONO2 "collar" occurs with ClONO2 VMRs bigger than before the winter because the HCl has been depleted by heterogeneous reactions. So, the bottom line is that the biggest values of ClONO2 also depend on the heterogeneous chemistry.

We added these details to the paragraph.

**1.16 Line 165–167: Now, I am confused. When switching off heterogenous chemistry, I would have expected MUCH higher ozone values (a few ppm). Am I misunderstanding something here completely or is there something going wrong here (e.g. the values at the x axis not being correct)? And the ozone values in the northern hemisphere seem to be suspiciously low in all cases. Values of below 0.3 ppm are normally never observed in this altitude range, even with ozone depletion. I have the impression that something is clearly going wrong here.**

Thank you for pointing this out. We had an error of one order of magnitude when converting the volume mixing ratios to ppmv. The fixed figures can be found in the revised version of the manuscript (and supplement). We also corrected the connected numbers in the text of the manuscript.

**1.17 Lines 168 and following: Why don't you compare to ozone from MIPAS here? Would somehow be consistent to the rest of the manuscript. I would also find it interesting not only to see the differences, but also the mean ozone profiles.**

We decided to show profiles of ozonesondes because the precision of the single (in-situ) measurement is better than for satellite measurements. We added the mean profiles as panels to Fig. 7 in the revised manuscript and the figures in the supplement.

**1.18  Lines 207–208: Isn't it a little bit over the top to argue with the Kolmogorov- Smirnov test here when it can be easily seen from the figures that the distributions are still different? I think that could be shortened.**

Zambri et al. (2021) use a similar approach as in this study, but compare the PDFs with the K-S test which is why we would like to keep this statement in the manuscript. We added to this sentence that they use a similar approach in the revised manuscript to make this clearer.

**1.19  Supplement: I think it wouldn't hurt to mention what species you are showing in the plots. In the moment, you can only deduce that from the figure numbers in the main manuscript given in the caption.**

We added the species name to the figure captions.

**2  Technical corrections**

**2.1  Line 124: Typo "cpmpared"**

Corrected.

---

## Author Comment (AC2)

Dear Referee,

Thank you very much for this detailed review of our manuscript. Your comments helped us to improve the manuscript and we hope that we could address all of your major concerns in the revised manuscript. Please find below our responses to each of your comments. Major changes include:

- We included WACCM in the manuscript title

- We added more information about the NAT parameterization in WACCM

- Some more information about the method how to calculate the max(d) values is provided in Sect. 2.2

- We added two new figures comparing MIPAS $HNO_3$ with WACCM as (1) a timeseries and (2) a scatter plot, see discussion below and in the revised manuscript

Thanks again for your review and on behalf of all authors,
Michael Weimer

**1 General Comments**

**1.1 The most important point I would like to address is the lack of comparison of the simulated NAT density with observations. Although this is the most important set screw for this study, no attempt is made to constrain the NAT density used in this modeling study with observational data. Satellite-based observations of NAT density are mentioned in the introduction but are not discussed further. The model results are compared to more than 10 years of trace gas observations. I assume that observations of NAT density are also available during this period, at least temporarily.**
– and –
The NAT density now recommended as an input parameter differs by a factor of 20 from the previous one. Although the simplifications of the model might not allow direct integration of the measured quantities, it is necessary to relate the model simulations to observations of the NAT density.

The previously used NAT number density of $10^{-2}\,\mathrm{cm^{-3}}$ was based on measurements by CALIOP, but as we show in this study, this value is too large to reflect

an effect of dentitrification comparable to the MIPAS trace gas measurements. We added some details about the NAT parametrization to Sect. 2.1 and also added some of the main simplification of the NAT scheme in WACCM: (1) NAT particles in one $100 \times 100\,\mathrm{km}^2$ grid box are assumed to have one size which is not the case in reality, (2) NAT particles in this scheme are not allowed to grow or decrease over time and (3) the whole amount of supersaturated gaseous HNO3 is transferred to NAT. Due to these simplifications, we think that this value should not be guided by measured values of the NAT number density. Nevertheless, we added some measured values of the NAT number density to put this value into context.

**1.2 The manuscript states that reduced NAT density in the model setup leads to better agreement between model simulations and observations. And this seems to be true for HNO3, ClONO2 and ozone. But why is this so? What are the underlying physical and chemical processes that are responsible? At least a rudimentary attempt should be made to explain this plausibly.**

A lower NAT density will lead to a vertical redistribution of HNO3 (lower HNO3 at high altitudes and higher HNO3 at lower altitudes). During early spring, HNO3 will be photolyzed and form NO2 which will combine with ClO to form ClONO2. ClO is responsible for the catalytic ozone depletion and deactivated through this previous reaction. Therefore, denitrification will influence HNO3, ClONO2 and ozone. We added this explanation at the start of Sect. 3.

**1.3 In general, a more detailed discussion of the simplifications of the model would be helpful to better classify the quality of the presented comparisons between model and observations in this study. For example: the statement that "... so observed NAT particle abundances may therefore not be the best guide for this parameter choice ..." (page 3, line 68) should be discussed in a bit more detail.**

We added a more detailed description of the NAT parameterization of WACCM in Sect. 2.1 and also put this statement into this context.

**2 Specific comments**

**2.1 Page 4/line 105: Although the term "maximum difference" might be common in the modeling community, it would be useful for a broader readership to see this concept presented and explained in a bit more detail.**

We added the equation how the maximum difference is calculated, now Eq. 1 in the revised manuscript.

**2.2 Page 5/line 121: "... Turning off all heterogeneous chemistry except for N2O5 + H2O ..." Why is this reaction not turned off?**

We are focusing on the effect of halogen chemistry and denitrification, which are highly temperature dependent. The N2O5+H2O reaction rate is nearly independent of temperature, happens on nearly all aerosols and does not directly affect the halogen chemistry, which is why we kept this reaction for noHetAll.5e-4. We added an explanation to the revised manuscript.

**2.3 Figure 1: The labels in the partial figures take up too much space (HetAll ....). This means that the actual content of the figure is not displayed optimally. Since the color coding of the simulations is the same in all subfigures, it could be placed separately next to the figures. The indication "max(d)" should of course remain in the figure.**

We made the labels annotations in each panel and put the legend below the figure.

**2.4 Figure 1: What does the indication "nprof:...." at the top of the figure mean?**

It's the number of profiles used to derive the PDFs of the panel. It depends on the size of the polar vortex and is different for each year and hemisphere. We added this explanation to the caption of Fig. 1.

**2.5 Figure 1 to 4: The volume mixing ration of nitric acid in the stratosphere varies with altitude, similar as ozone (Figure 5). In addition, sedimentation of PSC particles leads to a change in the height distribution of HNO3. In Figures 1 through 4, only altitude-independent concentration is given. Please explain briefly how this value is obtained. Would it not be possible, at least for one case of HNO3, to choose a similar representation as for ozone in Figure 5?**

We added a new figure showing the timeseries of vortex-averaged HNO3 profiles of MIPAS and all WACCM simulations, now Fig. 1 in the revised manuscript. In addition, we added a scatter plot including all altitudes which shows a concise correlation between WACCM and MIPAS in the case of HetAll.5e-4. This supports that we can use this large pressure range in the PDFs of the next figures.

**2.6 Page 5/line 124: "... In HetAll.1e-2, larger HNO3 values are more common in all panels compared to the other simulations..." Can you explain this result? Why does a higher density in the particle phase lead to a higher gas phase concentration of HNO3 and vice versa? What processes underlie this behavior?**

We added some statements about this to the revised manuscript: In Sect. 2.1 we explain now that a higher NAT number density leads to smaller particles, which then impacts the sedimentation velocity which is decreased. Therefore, a larger fraction of gaseous HNO3 remains at higher altitudes which can be also seen in the new Fig. 1 in the revised manuscript and which we also discuss there.

**2.7 Page 7, line 148: In the manuscript, the discrepancy between model and measurement is attributed to the lower accuracy of the MIPAS instrument. By how much was the accuracy of the measurement reduced compared to the measurements included in Figure 1? Can model processes be excluded for this discrepancy?**

We replaced "precision" by "accuracy" and added other possible reasons for the mismatch between MIPAS and WACCM for that year.

**2.8 Figure 5: The "interquantile range" is very difficult to see in the figure.**

We increased the line width of the interquartile range in this figure and also the figures in the supplement.

---

## Referee Report (RR1)

I am happy to accept the manuscript with a few very minor corrections. I will be fine with not seeing the next revised version.

All line numbers refer to the "tracked changes" manuscript.

**Comments to your replies (no action required except for 1.15)**

- 1.4 I acknowledge your decision not to use MLS data, but I think your argumentation in your reply is not valid. You are either showing vortex means (new Fig. 1), pdfs or a scatter plot based on the profiles inside the vortex (new Figs. 2, 3, 4, 5). That means that the statistical error (precision) of MLS or MIPAS data is negligible in your "final product", either because of the average of many profiles (Fig. 1) or because these errors cancel out in a pdf (or scatter plot) and don't change the pdf significantly (except maybe at the tails). That leaves the systematic error (accuracy). For HCl, this is for instance 0.2 ppb in the considered altitude range in the version 5 data, which is reasonably low (similar for other species). Consequently, MLS has been used in many studies for comparison to model data (I hope I don't have to give citations, you probably can easily come up with a list).

- 1.9: I am happy about the changes. I think Figures 1 and 2 are very valuable additions to the manuscript.

- 1.12: Fine with your reply. But a little bit surprised that there is no further reasoning behind choosing these winters.

- 1.15: Do I interpret you correctly that you would like to say that some of the chlorine originally in HCl at the start of the winter shows up in ClONO2 at the end of the winter? It might help to phrase it like this in the manuscript. In the moment, the formulation is not really to the point.

- 1.16 I am very happy that this was a simple conversion error.

- 1.17 Agreed and I acknowledge your decision not to show MIPAS data. But nevertheless, this would have been easy to add.

**Specific comments**

- 96–97, comment of other reviewer: I think the main reason why this reaction should be kept is that it also happens on the background aerosol and not only on PSCs, and is important for the NOx partitioning. Without this reaction, the chemistry in the model would be completely unrealistic. This is partly stated in your sentence, but could be made clearer.

- 166–168: If this sentence is meant to serve as a justification for using the pdfs, you can only argue with the years here and not with the pressure range. The pressure range stays the same in Fig. 2 and Fig. 3.

**Technical corrections**

- 46: "solid HNO3 of NAT" : "HNO3 contained in NAT particles"

- 48: "stadard" : "standard"

- 71: I would write "NAT threshold temperature", because this is not necessarily the formation temperature in reality (supersaturation).

- 90: Is "decrease" the right word? "shrink", "evaporate" etc.

- 125: "sigificant" : "significant"

- 134: "illustration" : "discussion"

- Fig. 1 font size could be a little larger. Very hard to read. For the labels on the y axis (pressure), I would prefer 40, 60 and 100 hPa instead of the exponential notation.

- 148: "deactivated through this mentioned reaction" sounds a little odd. "deactivated by the reaction into ClONO2" or something like this.

- 152: "decreasing" : Maybe "HNO3 VMR decreasing with time"?

- 155: Logic: "...do not get large enough for the simulated HNO3 to compare well..."

- Fig. 2 caption "corresonds" : "corresponds"

- 163: "concise correlation" : "compact correlation"

- 164: "show a larger spread" : "show a larger scatter"?

- 164-165: Formulation: Suggestion: "The simulation HetAll.1e-2 shows HNO3 VMRs that are too large compared to the observations because of NAT particles that are too small."

- 165: Split sentence: Start new sentence: "As already suggested by Fig. 1..."

- 165: Rephrase the sentence starting in 165 in a way similar to my suggestion 164-165.

- Fig. 7 : Font sizes are a bit small.

---

## Author Response (AR2)

Dear Dr. Khosrawi,

Thank you for your letter associated to our manuscript submitted to ACP. We attached our point-to-point responses to the latest referee report to the submission. Here you can find our responses to your concerns described in your letter. References added here can be found below this response.

*Re your questions 1 and 4:*

*1. For me also the answer why you cannot compare WACCM to MLS is not satisfiying. I compared ECHAM5/MESSy simulations to both instruments without encountering any problems (Khosrawi et al. 2017, 208). Both instruments are suited for comparisons with model data. Generally, I would say that the decision is rather made by which data set is available and maybe also to some extent which fits better to the model data. The comparisons in Khosrawi et al. (2018) showed that e.g the HNO3 distribution looks generally the same, but the absolute values are quite different, which I think will in case of your study change the results to some extent.*

*4. Since there are large differences between the instruments I doubt that you can find a general solution for the "best" NAT density value by just comparing with one instrument. There we are again back at comment 1, I am not convinced by your arguments that it is technically not possible to perform the same study with MLS data. You need better arguments why you only use one instrument and why only MIPAS.*

It is important to note here that the comparison with MIPAS in the manuscript is based on a sampling of the model at exactly the same locations and times as the MIPAS measurements. Therefore, this is

[Figure]

*Figure 1:* MLS and MIPAS HNO3 VMRs vs. N2O VMRs at high latitudes for both hemispheres during the local fall months in 2011. Data screening as recommended by the MLS documentation has been applied to the MLS data. The pressure range 60 to 75 hPa is shown.

based on single point comparisons, which means that precision of the instrument is crucial when applying our method. As stated e.g. by Piccolo and Dudhia (2007) and Sheese et al. (2016), the precision of MIPAS (about 0.2 ppbv or even lower) during the local spring months is substantially better than MLS (about 0.6 ppbv, see MLS data documentation , Livesey et al., 2022). In addition, the vertical resolution of MIPAS is about 3 km in the stratosphere which is higher than for MLS. MIPAS was able to measure profiles to latitudes up to 87 °N/S which is not possible with MLS. Figure 1 shows MLS and MIPAS $HNO_3$ vs. $N_2O$ for both hemispheres (70 to 82 °N/S) at around 68 hPa during local fall months and it can be seen that the scatter is considerably larger for MLS than for MIPAS. For all these reasons, we decided to compare our results with MIPAS. Note that MIPAS is not the only dataset used but we also use ozonesondes for comparison with the same message as the MIPAS comparisons, which supports the conclusions of our study.

*Re your question 2:*

*2. I cannot understand why one NAT density values needs to be used for both hemispheres. There are so many differences between the hemispheres in PSC occurrence and formation, would it not then be wiser to use different values and thus optimise these for the two hemispheres separately? At the moment it looks a bit like with every new comparison the NAT density value is adjusted (where we are back at my first comment), meaning for each (extreme) winter, satellite data set a new "best value" is defined.*

The "real world" NAT density and subsequent denitrification can vary significantly based on air-parcel history (i.e., the Lagrangian trajectory) of temperature magnitude and water-vapor & $HNO_3$ abundance. The "model" choice to use a "one" number representation is not to suggest that the current parameterization is going to be directly comparable to the observed representation of NAT particle density. The point of changing this parameter in a Eulerian CCM (i.e., like WACCM6) is to get a reasonable representation of the observed gas-phase $HNO_3$ distributions in both hemispheres. Getting the right abundance of $HNO_3$ (g) is necessary for adequately representing ozone loss. The previous choice of the "model" NAT density (0.01 particles $cm^{-3}$) greatly underestimated the denitrification impact in the Northern Hemisphere, especially in cold winters (e.g., 2011 and 2020). Using the new particle density value (5 x $10^{-4}$ particles $cm^{-3}$) we were able to improve the Northern Hemisphere $HNO_3$ abundance and subsequent ozone loss. It also slightly improved the Southern Hemisphere $HNO_3$ distribution. The Southern Hemisphere is less sensitive to the "model" NAT density parameter choice due to the long period of cold temperatures where near complete denitrification can occur.

*Re your question 3:*

*3. You provided some measured values of the NAT density which is important. However, how does your value fit into the range of measured values? This should also be shortly discussed. If it does not fit into the range of measured values you have a serious problem, because then it looks like you just find the perfect NAT density value to tune your model. By looking at the values, however, I had the feeling you where at the lower end of the measured values.*

We would like to emphasize here (as also stated in our responses to the referees) that the NAT density does not necessarily need to be comparable to measured values. As we discussed in the responses to the first review, the NAT density in the model parameterizes more than only the physical

quantity of a NAT density due to the gross simplifications in the NAT parameterization. It is a tuning parameter which can be used to lead to the goals: realistic global distributions of gaseous $HNO_3$. Nevertheless, we added a sentence to the conclusions about its correspondence to the measured values.

We hope we have addressed your concerns with these responses.

Yours sincerely and on behalf of all authors,

Michael Weimer

**References**

Livesey, Nathaniel J., William G. Read, Paul A. Wagner, Lucien Froidevaux, Michelle L. Santee, Michael J. Schwartz, Alyn Lambert, Luis F. Millán Valle, Hugh C. Pumphrey, Gloria L. Manney, Ryan A. Fuller, Robert F. Jarnot, Brian W. Knosp and Richard R. Lay: Version 5.0x Level 2 and 3 data quality and description document. https://mls.jpl.nasa.gov/data/v5-0_data_quality_document.pdf, 2022

Piccolo, C. and Dudhia, A.: Precision validation of MIPAS-Envisat products, Atmos. Chem. Phys., 7, 1915–1923, https://doi.org/10.5194/acp-7-1915-2007, 2007.

Sheese, P. E., Walker, K. A., Boone, C. D., McLinden, C. A., Bernath, P. F., Bourassa, A. E., Burrows, J. P., Degenstein, D. A., Funke, B., Fussen, D., Manney, G. L., Thomas McElroy, C., Murtagh, D., Randall, C. E., Raspollini, P., Rozanov, A., Russell, J. M., Suzuki, M., Shiotani, M., Urban, J., Von Clarmann, T., and Zawodny, J. M.: Validation of ACE-FTS version 3.5 NOy species profiles using correlative satellite measurements, Atmos. Meas. Tech., 9, 5781–5810, https://doi.org/10.5194/AMT-9-5781-2016, 2016.

Dear Ingo Wohltmann,

Thank you for the review of our revised manuscript. Please find below our responses to your comments.
Thanks again for your review and on behalf of all authors,
Michael Weimer

**1 Responses to major comments**

**1.1 1.4 I acknowledge your decision not to use MLS data, but I think your argumentation in your reply is not valid. You are either showing vortex means (new Fig. 1), pdfs or a scatter plot based on the profiles inside the vortex (new Figs. 2, 3, 4, 5). That means that the statistical error (precision) of MLS or MIPAS data is negligible in your "final product", either because of the average of many profiles (Fig. 1) or because these errors cancel out in a pdf (or scatter plot) and don't change the pdf significantly (except maybe at the tails). That leaves the systematic error (accuracy). For HCl, this is for instance 0.2 ppb in the considered altitude range in the version 5 data, which is reasonably low (similar for other species). Consequently, MLS has been used in many studies for comparison to model data (I hope I don't have to give citations, you probably can easily come up with a list).**

The precision of individual measurements are key to the distributions that are of interest here. MLS data distributions are compared to MIPAS in a separate letter to the editor and shown to be less precise.

**1.2 1.15: Do I interpret you correctly that you would like to say that some of the chlorine originally in HCl at the start of the winter shows up in ClONO2 at the end of the winter? It might help to phrase it like this in the manuscript. In the moment, the formulation is not really to the point.**

Thank you, we reformulated this sentence.

**2 Specific comments**

(Please note that the line numbers refer to the "Author's Track Changes")

**2.1 96-97, comment of other reviewer: I think the main reason why this reaction should be kept is that it also happens on the background aerosol and not only on PSCs, and is important for the NOx partitioning. Without this reaction, the chemistry in the model would be completely unrealistic. This is partly stated in your sentence, but could be made clearer.**

Thank you for this comment. We added this explanation to the sentence.

**2.2 166-168: If this sentence is meant to serve as a justification for using the pdfs, you can only argue with the years here and not with the pressure range. The pressure range stays the same in Fig. 2 and Fig. 3.**

From our point of view, Figure 2 demonstrates that there is a correlation between WACCM and MIPAS HNO3 when using the recommended NAT density. As stated in the comment, the same pressure ranges are used which is why we think that this also demonstrates that we can use the whole pressure range in the following figures.

**3 Technical corrections**

**3.1 46: "solid HNO3 of NAT" : "HNO3 contained in NAT particles"**

Corrected.

**3.2 48: "stadard" : "standard"**

Corrected.

**3.3 71: I would write "NAT threshold temperature", because this is not necessarily the formation temperature in reality (supersaturation).**

Corrected.

**3.4 90: Is "decrease" the right word? "shrink", "evaporate" etc.**

We corrected this to "change their size over time" instead of "grow or decrease over time".

**3.5 125: "sigificant" : "significant"**

Corrected.

**3.6 134: "illustration" : "discussion"**

Corrected.

**3.7 Fig. 1 font size could be a little larger. Very hard to read. For the labels on the y axis (pressure), I would prefer 40, 60 and 100 hPa instead of the exponential notation.**

We adapted the figure accordingly.

**3.8 148: "deactivated through this mentioned reaction" sounds a little odd. "deactivated by the reaction into ClONO2" or something like this.**

Corrected.

**3.9 152: "decreasing" : Maybe "HNO3 VMR decreasing with time"?**

Corrected.

**3.10 155: Logic: "...do not get large enough for the simulated HNO3 to compare well..."**

Corrected.

**3.11 Fig. 2 caption "corresonds" : "corresponds"**

Corrected.

**3.12 163: "concise correlation" : "compact correlation"**

Corrected.

**3.13 164: "show a larger spread" : "show a larger scatter"?**

Corrected.

**3.14 164-165: Formulation: Suggestion: "The simulation HetAll.1e-2 shows HNO3 VMRs that are too large compared to the observations because of NAT particles that are too small."**

Corrected.

**3.15 165: Split sentence: Start new sentence: "As already suggested by Fig.1..."**

Corrected.

**3.16 165: Rephrase the sentence starting in 165 in a way similar to my suggestion 164-165.**

We rephrased the sentence.

**3.17 Fig. 7 : Font sizes are a bit small.**

We increased the font sizes in the figure.